# Subjective affective experience under threat is shaped by environmental affordances

**Song Qi** [1], **Dylan M. Nielson**[1], **Daniele Marcotulli**[2], **Daniel S. Pine**[1], **Argyris Stringaris**[3,4] *

**1** Emotion and Development Branch, National Institute of Mental Health Intramural Research Program, National Institute of Mental Health, Bethesda, Maryland, United States of America, **2** Section of Child and Adolescent Neuropsychiatry, Department of Public Health and Pediatric Sciences, University of Turin, Turin, Italy, **3** Divisions of Psychiatry and Psychology and Language Science, University College London, London, United Kingdom, **4** 1st Dept Psychiatry, National and Kapodistrian University of Athens, Athens, Greece

* a.stringaris@ucl.ac.uk

**Data Availability Statement:** Data and codes for the analysis can be found at our OSF registration page: https://osf.io/y5qv4/.

**Funding:** The work was supported by NIMH Intramural Research Program Project MH002781.

## Abstract

In this pre-registered study, we ask how people's emotional responses under threat may be causally affected by what is available to them in the environment, i.e. environmental affordances. For this purpose, we introduce a novel behavioral paradigm using horror movie stimuli to simulate threats. The study illustrates that affordances, specifically items present in the environment, are instrumental in modulating both behavioral choices (approach or avoidance) and emotional expressions of anger and fear. We found that, approach-related resources, such as possession of a weapon, heightened anger and the propensity to confront the threat. This underscores the influence of environmental affordances on emotional regulation and supports a theoretical framework that connects instrumental motives with the variability of emotional and behavioral responses based on affordances. The research, while innovative, recognizes the constraints of simulated threats and controlled settings, suggesting avenues for future exploration in more naturalistic environments.

## Introduction

Fear and anger are subjective experiences evoked in response to threat. However, it remains unclear whether in any given threat situation, fear or anger will be predominantly experienced. Here we use a novel experimental approach to demonstrate how environmental affordances causally and differentially impact anger and fear levels.

Humans, like other animals, deploy various responses when they encounter threats [1–3]. Emotions are programmes that coordinate sets of responses including a cognitive (e.g., attention bias) and physiological (e.g., vasoconstriction) changes [4]. Subjective affective experiences, often referred to as feelings [5], are one such response to threat. Fear and anger are prototypical feelings evoked by threats and associated with fight or flight reactions [6], i.e., the set of responses necessary to either escape or combat threats. Highlighting their public health importance, intense feelings of anger and fear also define a host of very common mental disorders [7].

We have a limited understanding of factors determining why we experience either fear or anger in any given threat situation. Typically, factors are identified through studies on inter-

The funders had no role in study design, data collection and analysis, decision to publish, or preparation of the manuscript.

**Competing interests:** The authors have declared that no competing interests exist.

individual differences—the extent to which differences in trait anger or fear relate to people's respose to a threatening situation. This approach has yielded important results about how people's life histories [8], cognitive biases [9] and affective traits [10, 11] influence how they respond to threats. Yet inter-individual difference alone does not encompass all of factors shaping emotional responses.

Identifying factors that influence our response to threats has several important implications. First, understanding the interplay between aspects of the environment and feelings can help situate subjective affective experience in contemporary emotion theory. Indeed, many theories emphasise the challenges with understanding the role of subjective experience in emotional responses [4, 12]. Second, understanding the effects of the environment on emotional experience can clarify how psychopathology is maintained and expressed. Environments may amplify feelings of fear or anger into dysfunctional traits. Modifying such environments or their perception could therefore prove helpful therapeutically.

We study environments under the term Affordance, which refers to resources offered by environment to an organism [13]. Consideration of environmental affordances primarily derives from perception psychology; yet the term also applies to the context of threat. Objects in the environment can offer different affordances with different utilities. We predict that the same stimulus appearing with different affordances invokes different feelings in people. Specifically, we predict that people faced with threat experience disproportinate levels of anger when means in the environment support fighting but disproportionate levels of fear when means in the environment support escape.

We base predictions about environmental affordances on considerations about the instrumentality of emotional experience. Some approaches to emotion emphasize hedonic emotion regulation [14]. In contrast, instrumental emotion regulation offers an empirical and theoretical grounding for our predictions [15]. The central tenet of instrumental emotion regulation is that people's experience with emotions not only reflects hedonic (pleasure-related) value, but also the utility of an emotion, the perceived advantage it provides. In summary, there is indeed evidence that prior to engaging in conflict, people are more likely to choose situations that are likely to elicit anger rather than neutral emotions [16].

To test these predictions, we developed a novel experimental framework in which each threat stimulus is presented multiple times with different affordances, in the context of horror movie clips. Narrative media, such as movies and stories, are powerful tools for examining action and agency. They provide a rich context for understanding how individuals perceive and engage with their environments [17, 18]. This also isolates causal effects of affordances on subjective affective experience. In our experiments, we manipulated both the resources available to fight (fight affordances) and the means of escape (escape affordances), in the form of written descriptions. Affordances, traditionally understood as opportunities for action provided by the environment, also beyond physical objects to include written descriptions that can evoke emotional responses [19–21]. We label fight affordances in which a weapon or weapon-like tool are available as the "weapon condition", as opposed to the "non-weapon condition", where an available tool is not suitable for fighting. Escape affordances in which escape is more difficult are referred to as the "non-evadable condition" as opposed to the "evadable condition." Clips from horror movies were used as the stimulus, while participants were asked to rate their emotional responses and decisions under the movie scenarios.

We used two steps to support the spirit of open science and to ensure replicability. We first conducted a pilot study and used its results to specify our hypotheses and estimate effect sizes for the purposes of power calculations, which we pre-registered. In the second step, we collected data on n = 250 individuals to test these hypotheses using both a frequentist and a Bayesian approach. We find that environmental affordances in the form of items available to fight or

to escape when under threat, strongly influence both decisions to fight or to escape and emotions expressed under threat. Participants experience more anger in threatening situations while choosing to fight during fight affordances; they also experience more fear while choosing to escape under escape affordances. In other words, we demonstrate that anger expression is heightened with the alignment between environmental affordances and the actions chosen by the individual, such that one displays more anger when choosing to fight under the weapon condition, compared to the non-weapon condition.

## Methods

### Participants

The study population consisted of ordinary, non-selected adults aged 18 years or older. Participants were recruited through Amazon Mechanical Turk (MTurk) and were eligible to participate if they met the following criteria: (1) currently residing in the United States, (2) having an average HIT (Human Intelligence Task) approval rate of 95% or higher, and (3) having completed a minimum of 1000 HITs.

We first conducted 2 plot studies, one of them with only the fight affordances, and the other with both fight affordances and escape affordances. Due to technical difficulties, we were not able to save the full data from the pilot studies, and thus decided not to include them in the formal analysis. We used the design from the pilot study to conduct a power analysis.

We aimed to recruit a total of 250 participants to ensure adequate statistical power (>80%) for the analyses. Power analysis was conducted for both the frequentist and Bayesian analysis, and a sample size of 250 was selected for both standards to reach adequate power. Data collection was conducted using the Pavlovia.org platform. Participants first completed a brief demographic questionnaire and were then provided with detailed instructions for the main experimental task. Upon completion of the task, participants were debriefed about the study's objectives and their MTurk Worker ID was collected for compensation purposes.

Participants: A total of 250 participants were recruited for the study (155 male, 95 female) with an age range of 18 to 60 years (M = 30.9, SD = 9.42). The ratio of approved HITs vs. total completions is 92.4%. In terms of self-rated SES (Social-economic status), 12.0% rated themselves as "high"; 8.4% as "above middle"; 20.8% as "middle"; 34.2% as "below middle"; 24.6% as "low".

Participants' Worker IDs were initially matched with the corresponding MTurk records to confirm HIT approval and verify eligibility. This is only done to check the status of task completion, and no data were analyzed at this stage. All personal identifiers, including the Mturk Worker's IDs were subsequently removed before entering analysis.

The study protocol was approved by the NIH Institutional Review Board Operations (NIH-IRBO), Reference Number 528098. All participants provided written informed consent online before participating in the study. The collection of data started on 02/05/2021, and ended on 08/10/2023.

By employing these inclusion criteria and data verification methods, we aimed to ensure the accuracy and reliability of the findings from this study. However, it is important to note that the MTurk population may not be fully representative of the general population, and thus the external validity of the results may be limited.

Data and codes for the analysis can be found at our OSF registration page: https://osf.io/y5qv4/.

### Experimental procedure

The basic elements of our experiments are presented below. In the beginning of each trial, participants watch a short video (10 seconds) excerpted from a horror movie scene (Threat). They

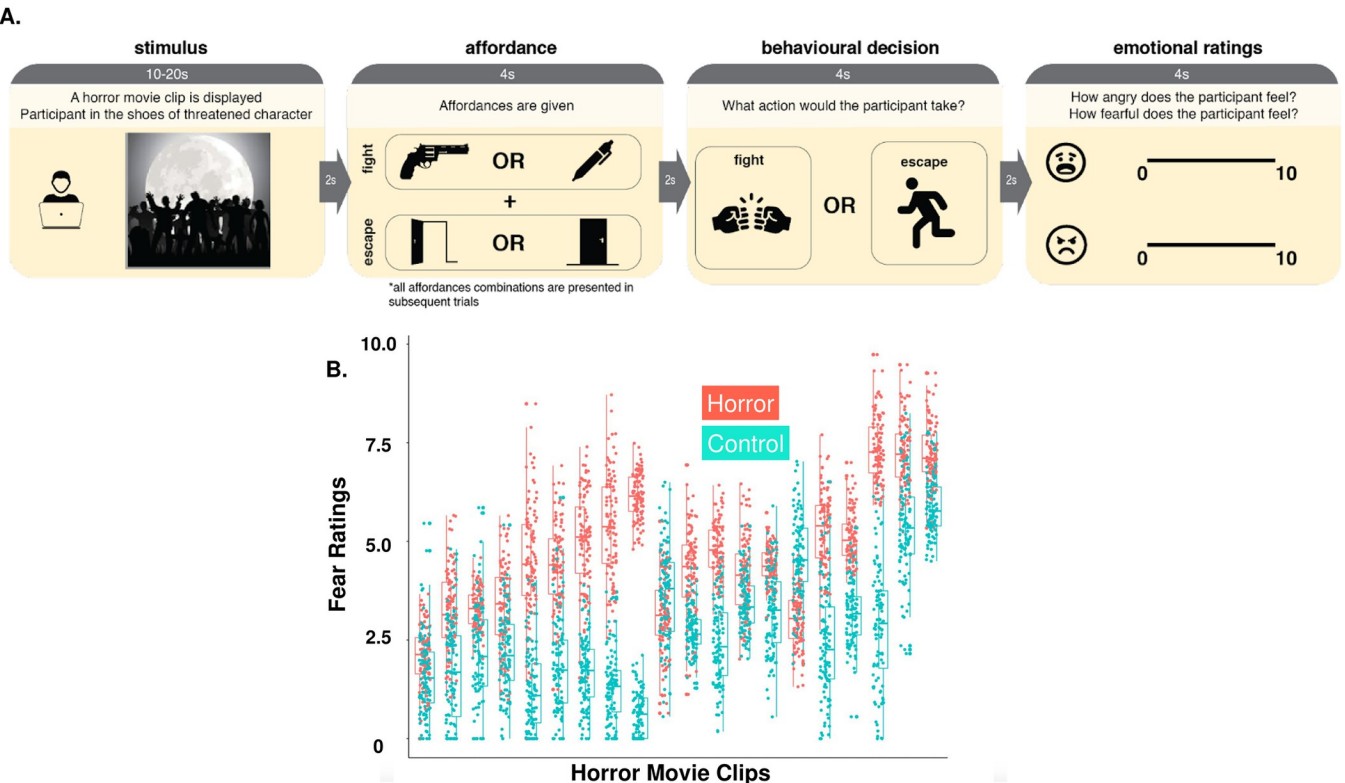

**Fig 1.** (A) Experimental Procedures. During each trial, the participant is first presented with a horror movie clip, where the protagonist needs to make an approach-avoidance decision for survival. The participant is then presented with information regarding the environmental affordances, including the fight affordance (e.g., having tools that can be used as a weapon or not) and the escape affordance (e.g., whether the escape route is blocked or not). With this affordance information, participants make their survival decisions and rate their emotional responses. (B) Fearfulness rating for the sample movies. Clips from the horror scenes are rated more fearful than the non-horror scenes. This offers evidence that the survival decisions were prompted out of a genuine feeling of fear. Please refer to S1 File.

are instructed to put themselves in the shoes of the threatened characters in the video and make choices that they consider to be best. They are instructed that the "best" choices would maximally increase the characters' chances of survival. Note that the outcomes of the movies (hence the result of their decisions) are not revealed at the end of the trials. This help enhance participants' impression that their choices can actually shape the development of the events in the movie and minimizes the possibility that previous outcomes would impact later choices. A graphical presentation of the schematic can be found in Fig 1.

Each threat presentation is accompanied by the depiction of both the fight affordance (fight facilitated or fight blocked) and the escape affordance (escape facilitated or escape blocked condition). This is a within-subjects factorial design with 2 factors (fight affordance X escape affordance) with 2 levels each (relevant and control). Each participant will be asked to watch a total number of 40 video clips paired with the 4 affordance levels. A list of affordances can be found in S1 File.

After the presentation of the movie clips, participants are asked to make a choice between approach behavior (fight through the situation) and avoidance behavior (escape from the situation). Participants can choose either "fight" or "escape" on each decision trial. To simulate realistic environmental affordances, the "fight" or "escape" choices were expressed in a descriptive language like "Use the shovel to fight back". The choice was dichotomous and encoded as 1 and 0 respectively.

Finally, after making their choices, participants are asked to rate their emotions, by being asked "How fearful do you feel?" and "How angry do you feel?" by moving a cursor on the screen to match a number that reflects their relative rating of the question. A continuous scale bar ranging from 1 (not angry/fearful at all) to 10 (very angry/fearful) was used for the rating. Participants can freely move the cursor on the scale to indicate their rating.

**Randomization.** We use block randomization, where each participant is randomly assigned to a seed-generated order of video presentations. The presented threat videos are the same across participants–only the orders of presentation were altered. The purpose of the randomization is for counterbalancing, and to account for the familiarity brought by the same movies.

## Data analysis

**Hypothesis 1 (H1):** Environmental affordance impacts behaviors under threat. In the weapon condition trials we expected that participants would be more likely to choose to fight rather than flee compared to the non-weapon conditions (regardless of escape affordance). Similarly, in evadable trials, we expected that participants would be more likely to choose to flee rather than to fight, compared to the non-evadable trials (regardless of fight affordance) this corresponds to H1 in the following: https://osf.io/kxnyp for fight affordance, https://osf.io/sjcwu for escape affordance.

To test H1, we conducted a $\chi 2$ -test comparing the percentage of participants' fight choices out weapon vs. non-weapon trials, and in evadable vs.non-evadable trials. The inference criterion was a $p \leq 0.05$ in each of the two $\chi 2$ -tests.

**Hypothesis 2 (H2):** Environmental affordance impacts the experience of emotion under threat. We expected that participants would rate themselves as angrier in weapon trials compared to non-weapon trials and that participants would rate themselves as more fearful in evadable trials compared to non-evadable trials. This corresponds to H2 and H3 in the following: https://osf.io/kxnyp for fight affordance, https://osf.io/sjcwu for escape affordance.

To test H2, we used two methods, frequentist, and Bayesian. First, we applied repeated-measures, two-way ANOVA (Affordance X Choice) for participants' fear ratings and anger ratings. The coefficient of interest for H2 is the main effect of the affordance. We also looked at the interaction term between affordance and choice. The inference criterion was a $p \leq 0.05$ in the coefficient estimates for the two ANOVA models.

Second, to address the crossed-random effects structure in our design, we constructed a Bayesian mixed effects model to look at the relationship between emotional ratings and decisions. For the Bayesian analysis, we used the framework for Multilevel linear models (MLM). By recursively modifying the model structure and transforming random slopes to complex random intercepts (CRIs), the method reduces type-one error inflation and produces reliable model structures. While the standard MLM is usually fitted in a frequentist framework, if one tries to include the maximal varying structure (as in the case for the affordance data), the model tends either not to converge, or to give aberrant estimations of the correlation between varying effects. On the other hand, fitting the maximal varying effect structure has been explicitly recommended, and can be done with a Bayesian framework without non-convergence problems.

For our models, the null hypothesis postulates that the estimates of a corresponding item in the model is no different than 0. The alternative hypothesis postulates that there is enough evidence to conclude the estimates to be non-zero. We report the estimates, 95% credibility interval, and corresponding Bayes factors with regards to the null. While a Bayes factor = 1 indicates no evidence for the alternative hypothesis, Bayes factor between 1 and 3 indicates

anecdotal evidence for the alternative hypothesis; Bayes factor between 3 and 10 indicates moderate evidence for the alternative hypothesis; Bayes factor $> 10$ indicates strong evidence for the alternative hypothesis [22].

Bayesian analysis was performed through the brms computation package [23].

## Results

*Hypothesis 1*: to test hypothesis 1 (H1), that environmental affordance impacts behaviors under threat, we examined the participants' decisions to fight or flee.

On each trial, 4 possible combinations of affordances exist:

1. weapon + evadable condition

2. non-weapon + evadable condition

3. weapon + non-evadable condition

4. non-weapon + non-evadable condition

In the analysis to test hypothesis 1 (H1), we collapsed across escape affordances when analysing fight affordances and vice-versa. According to our H1, people would choose to fight more in weapon conditions (1 and 3) compared to non-weapon conditions (2 and 4). People would choose to escape more in evadable conditions (1 and 2) compared to non-evadable conditions (3 and 4).

Overal, participants chose to escape more than fight, given the seemingly non-opposable threat presented by the horror movie clips. We compared participant's percentage of fight choices in weapon conditions (1 and 3) vs. non weapon conditions (2 and 4). As predicted, participants chose to fight more during the weapon conditions (fight chosen: 41.54%, 95% CI = [38.68%, 44.40%]), compared to non-weapon conditions (fight chosen: 16.84%, 95% CI = [14.64%, 19.04%]; $\chi 2 = 1544.59, p < 0.001$), and escape more in the evadable conditions (escape chosen: 80.37%, 95% CI = [78.27%, 82.47%]) compared to the non-evadable conditions (escape chosen: 66.37%, 95% CI = [63.56%, 69.18%]; $\chi 2 = 486.07, p < 0.001$) (Fig 2(A)). This suggests that in accordance with our theory, people are more likely to choose actions that are favored by the environmental affordances and confirms hypothesis 1 (H1).

*Hypothesis 2*: To investigate hypothesis 2 (H2), that environmental affordance impacts the experience of emotion under threat, we applied repeated-measures, two-way ANOVA (Affordance X Choice) for both the anger emotional ratings and fear emotional ratings in the experiment. In these analyses we again collapsed across escape affordances when analysing fight affordances and vice-versa.

### More anger when having a weapon; but only when the decision is to fight

When looking at anger ratings accompanying their behavior choices, we found that participants rated themselves as angrier in the weapon conditions (1 and 3; mean anger rating [95% CI] = 4.59 [4.28, 4.90]), compared to the non-weapon conditions (2 and 4; mean anger rating [95% CI] = 3.71 [3.41, 4.01]). As shown in Fig 2(B), the difference in anger ratings can be observed on the individual level (mean difference [95% CI] = 0.88 [0.27, 1.49]).

By applying a repeated-measures, two-way ANOVA (Affordance X Choice) for the anger ratings, we observed both a main effect of affordance ($F_{2.60, 319.99} = 93.588, p < 0.001$) and an interaction between affordance and choice ($F_{2.73, 355.74} = 37.760, p < 0.001$). Post hoc comparisons showed that anger ratings were higher in the weapon conditions compared to the non-weapon conditions, only when the choice is to fight (anger rating in weapon conditions [95%

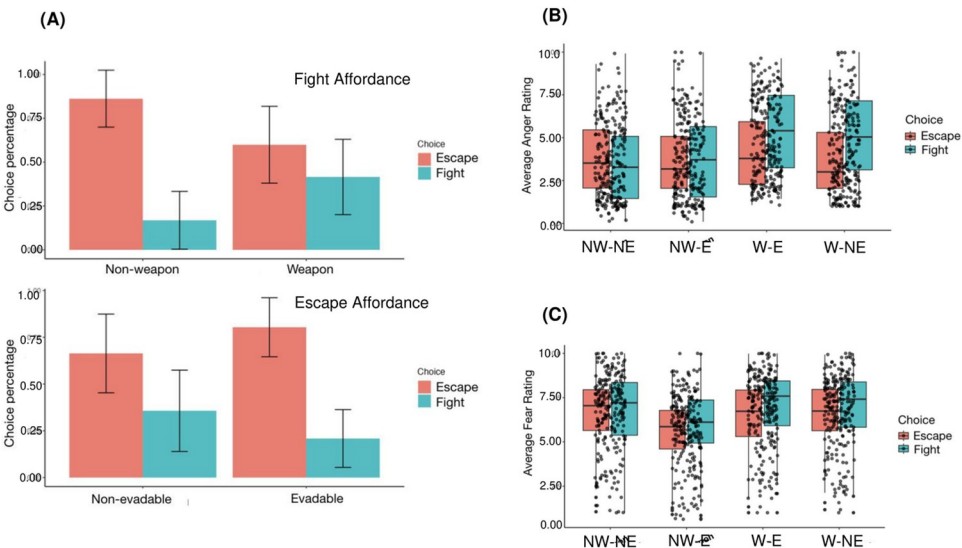

**Fig 2.** (A) Percentages of choices under different fight and escape affordances. Participants chose to fight more under the weapon condition, compared to the non-weapon condition. Similarly, they chose to flee more under the evadable condition, compared to the non-evadable condition. (B) Anger ratings with jitter points as individual averages. Lines on boxplots indicate 95% Cis. (C) Fear ratings with jitter points as individual averages. Lines on boxplots indicate 95% Cis. NW-NE non-weapon and non-evadable; NW-E: non-weapon and evadable; W-NE: weapon and non-evadable; W-E; weapon and evadable.

CI] = 5.24 [4.83, 5.65]; anger rating in non-weapon conditions [95% CI] = 3.69 [3.30, 4.09]; $T_{247}$ = 18.03, p < 0.001).

Notably, there were no significant differences in anger ratings between evadable (mean anger rating [95% CI] = 4.07 [1.72, 6.42]) and non-evadable (mean anger rating [95% CI] = 4.23 [1.90, 6.56]) conditions ($T_{247}$ = 2.10, p = 0.22).

## More fear when the escape is harder; but only when the decision is to escape

By applying a repeated-measures, two-way ANOVA (Affordance X Choice) for the fear ratings, we observed both a main effect of affordance ($F_{2.48, 305.47}$ = 130.391, p < 0.001) and an interaction between affordance and choice ($F_{2.50, 307.68}$ = 13.177, p = 0.03). Post hoc comparisons showed that fear ratings were higher in the non-evadable conditions compared to the evadable conditions, when the choice is to escape (fear rating in escapable conditions [95% CI] = 5.95 [5.60, 6.30]; fear rating in non-escapable conditions [95% CI] = 6.51 [6.16, 6.87]; $T_{247}$ = 9.68, p < 0.001).

## Variances explained by inter- and intra- individual differences

We also calculated the variances explained by inter-individual (between-group) differences and intra-individual (within-group) differences, expressed as the ratio of sum of squares: SS_within/SS_total and SS_between/SS_total. For anger ratings, the inter-individual variance ratio is 0.612, while the intra-individual variance ratio is 0.388; for fear ratings, the inter-individual variance ratio is 0.618, while the intra-individual variance ratio is 0.381. This suggests that participant's subjective expression of emotions are not foverned by differences in their traits, but rather largely influenced by environmental affordances.

To further address the effects brought by individual traits, we ran two additional ANOVA anlsysis on the same data set: 1) Anger rating X Anger-approach scores; 2) Fear rating X Trait

anxiety scores. In both cases, we found no significant effect from the trait scores (Anger-approach: $F_{1, 228} = 2.125$, p = 0.15; Trait anxiety: $F_{1, 228} = 3.027$, p = 0.08), and no interaction between affordances and trait scores (Anger-approach X Fight Affordance: $F_{1, 228} = 1.116$, p = 0.29; Trait anxiety X Escape Affordance: $F_{1, 228} = 1.588$, p = 0.21). We did not find evidence for an interaction between affordances and trait level factors on fear or anger ratios.

## Bayesian statistics

A Bayesian multilevel linear model was fitted using the brm function from the *brms* package in R. Two models were fitted for anger ratings and fear ratings respectively, with anger and fear predicted by participants' choice, affordance conditions, their interactions, and all the related random structures. The full details of the model can be found in S1 File.

In the anger rating model, as shown in Fig 3(A), the Bayesian multilinear model revealed a significant interaction between fight Affordance and Choices on anger rating (Estimate = 1.47, 95% CI [1.23, 1.72]; BF against null model without interaction = 22.49). This suggests that the effect of affordance on anger rating depends on the choices participants made–consistent with our findings in the ANOVA analysis.

The estimate of 1.47, with a BF of 22.49 indicates strong evidence for a large effect: when participants are in the weapon condition and make the decision to fight, the anger rating changes by 1.47. With the rating ranging from 0 to 10, it represents about 15% of the total scale in the anger rating.

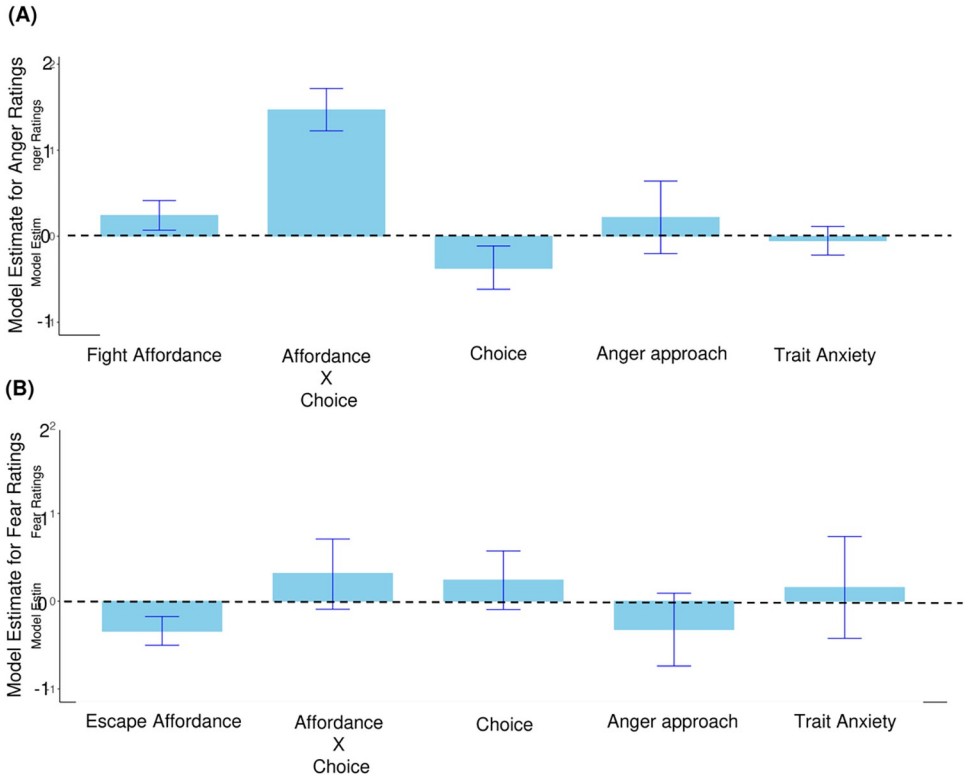

**Fig 3.** (A): Fixed effects from the Bayesian multilevel linear model predicting anger ratings. Consistent with the interaction term in the ANOVA anlalysis, there is strong evidence for a large effect from Affordance X Choice (Estimate = 1.47, 95% CI [1.23, 1.72]); (B): Fixed effects from the Bayesian multilevel linear model predicting fear ratings. With no reliable effect from either affordance, choice or trait anxiety themselves, we found a small effect from Affordance X Choice (Estimate = 0.32, 95%CI [0.09, 0.71]).

For the main effect of Choice we found moderate evidence for a medium effect (Estimate = -0.37, 95%CI [-0.62,-0.11]; BF against null model without main effect of choice = 4.11).

For the main effect of Affordance we found strong evidence for a medium effect (Estimate = 0.24, 95%CI [0.07, .41]; BF against null model without main effect of Affordance = 10.40).

In the fear rating model, as shown in Fig 3(B), there was moderate evidence for a medium effect of the interaction term between Affordance and Choices on fear ratings (Estimate = 0.32, 95%CI [0.09, 0.71]; BF against null model without the interaction: 7.27). Unlike the ANOVA analysis, there were no reliable main effects from either Choice (Estimate = 0.24, 95%CI [-0.09, 0.57]; BF against null model without the main effect of Choice: 2.91) or Affordance (Estimate = -0.34, 95%CI [-0.50, -0.17]; BF against null model without the main effect of Affordance: 3.10). Both the interaction and main effects are either unreliable or small, and this seems to be resulting from the weaker effect of affordance on fear, compared to anger ratings.

## Discussion

This paper reveals environmental affordances play a crucial role in decision-making and emotional experiences under threat. Our findings revealed that affordances, in the form of the availability of weapons and evadability of a situation, significantly influenced the choices and emotions of individuals under threat, providing support for Hypothesis 1 (H1) and Hypothesis 2 (H2). These results contribute to our understanding of the intricate relationship between environmental affordances, decision-making, and emotions in contexts of threat.

Our analysis of H1 showed that participants were more likely to engage in a fight when a weapon was available and attempt escape when the situation was evadable. This aligns with affordance theory's assertion that human behavior adapts according to the opportunities and constraints of the environment [13, 24]. Our findings also extend prior research on the influence of situational factors on decision-making [25, 26] by specifically emphasizing the importance of environmental affordances in guiding individuals' choices under threat.

In relation to H2, our findings indicated that the presence of environmental affordances impacted participants' emotional experiences under threat. We observed that participants reported feeling angrier when a weapon was available but only when they chose to fight. This finding is consistent with the appraisal theory of emotion [27], suggesting that emotions arise from cognitive appraisals of situational factors, like the presence of a weapon in this instance. Conversely, participants reported feeling more fearful when the situation was non-evadable, aligning with previous research that highlights the heightened perception of threat when escape options are limited [28, 29].

Our findings challenge the notion that affective experiences are explained by inter-individual differences. Indeed, in the frequentist analysis, 60% of variance was explained by inter-individual differences in predicting both anger and fear ratings. The lack of trait dominance is also shown by the Bayesian models. The impact of environmental affordances on both decision-making and emotional experience underscores the importance of considering the interplay between individual traits and environmental factors when studying affect and behavior.

Our results not only have implications for emotion theory, but also for clinical psychology, psychiatry, and public health. In terms of emotion theory, our findings lend substantial evidence to the idea that subjective affective experiences involve computational processes of environmental contingencies [30, 31]. In clinical psychology and psychiatry, our results suggest that changes in subjective affective experiences, which are central to psychological well-being and psychiatric disorders [32–34], could be achieved by modifying affordances and individuals' perceptions of their environment. Additionally, our findings may imply that the presence

of weapons or other indicators of potential violent action could increase risk not only through their availability [35, 36], but also through the subjective affective experiences they induce.

We acknowledge the limitations of our study, including the less variable escape affordances, compared to weapon affordances. While fight affordances are presented in the form of different items, escape affordances are shown only as either open exits or partially blocked exits. This could have resulted in less engagement with the escape affordances, and contributed to the lack of main effects in our Bayesian analysis. On the other hand, the use of hypothetical horror clips only allowed us to examine the immediate impact of environmental affordances on decision-making and emotional experiences. In addition, our affordances were binary in nature, and future experiments should use parametrically varied affordances to demonstrate dose-response effects. There is a lack of sensitivity in our design to account for the variance of efficacy of affordances in the "weapon" conditions, where one individual might find a particular item more/less useful against specific adversaries shown in the movies. However, in both our initial instructions and in training sessions, the participants were asked to focus on the emotional experience and intuitive reactions towards the threatening situations displayed in the movies, rather than finding a rational solution to wipe out the super-natural threats. Therefore, the subjective efficacy of the items will matter less in this scenario. If anything, this unaccounted-for variance would have decreased our ability to detect significant differences between conditions.

In conclusion, our investigation has demonstrated that environmental affordances drive both decision-making and emotional expressions in threatening situations. Notably, changes in emotions occur congruently when environments favor the decisions. These results provide a window into the role of environmental affordances in adaptive decisions towards threat. Future studies could parametrically vary affordances, develop more accurate computational models accounting for this mechanism, and expand focus to societal level phenomena.

## Supporting information

**S1 File. This file contains information regarding the sample size rationale, standalone rating task, Bayesian models, supplementary stimulus, pilot study, and the supplementary discussions.**
(PDF)

## Acknowledgments

We thank Patrizia Pezzoli for helpful comments on the manuscript.

## Author Contributions

**Conceptualization:** Song Qi, Dylan M. Nielson, Argyris Stringaris.

**Data curation:** Song Qi.

**Formal analysis:** Song Qi, Dylan M. Nielson.

**Funding acquisition:** Argyris Stringaris.

**Investigation:** Song Qi.

**Methodology:** Song Qi, Dylan M. Nielson, Argyris Stringaris.

**Project administration:** Song Qi, Argyris Stringaris.

**Resources:** Song Qi, Argyris Stringaris.

**Software:** Song Qi.

**Supervision:** Daniel S. Pine, Argyris Stringaris.

**Validation:** Song Qi.

**Visualization:** Song Qi, Daniele Marcotulli.

**Writing – original draft:** Song Qi.

**Writing – review & editing:** Song Qi, Dylan M. Nielson, Daniele Marcotulli, Daniel S. Pine, Argyris Stringaris.

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
