## [Decision Letter · Decision Letter 0]

27 Feb 2024

PONE-D-23-40401Subjective Affective Experience under threat is shaped by environmental affordancesPLOS ONE

Dear Dr. Qi,

Thank you for submitting your manuscript to PLOS ONE. After careful consideration, we feel that it has merit but does not fully meet PLOS ONE’s publication criteria as it currently stands. Therefore, we invite you to submit a revised version of the manuscript that addresses the points raised during the review process.

Three Reviewers have evaluated the manuscript, providing mixed revisions. I think that the manuscript is well written and the experimental procedure is inventive and has merits, but I agree that some shortcomings are present that prevent the reader to understand what exactly has been demonstrated by the study. I add two concerns of mine to Reviewers' comments:  - how affordances have been operationalized. As far as I understood, "affordances" were written words or maybe icons (judging by figure 1..? this should be clarified) - justification and possibly references should be provided to sustain that those can be properly considered affordances; according to Gibson, affordances are a matter of perception more than cognition; is a written word or a symbolic icon an affordance? To what extent?  - by using horror movies, this is basically a media psychology research, yet media psychology literature is absent. Immersion and understanding of narrative contents should be taken into consideration - personally I would make different choices if I had to face the mutant monster from "The Thing" or a person possessed by a demon... is a gun effective against both...? Are all weapons the same (e.g., close contact vs. distance)? Research on video games and interactive narrative may be useful to respond I suggest Authors to respond to Reviewers' comments taking into account that not all the revisions were supportive of the manuscript. I would also suggest to tone down interpretation of the results and improve limitations section of the discussion.

We look forward to receiving your revised manuscript.

Kind regards,

Stefano Triberti, Ph.D.

Academic Editor

PLOS ONE

2. Thank you for stating the following financial disclosure: "The work was supported by NIMH Intramural Research Program Project MH002781."

3. We note that Figure 1 in your submission contain copyrighted images. All PLOS content is published under the Creative Commons Attribution License (CC BY 4.0), which means that the manuscript, images, and Supporting Information files will be freely available online, and any third party is permitted to access, download, copy, distribute, and use these materials in any way, even commercially, with proper attribution. For more information, see our copyright guidelines: http://journals.plos.org/plosone/s/licenses-and-copyright.

Reviewers' comments:

Reviewer's Responses to Questions

**Comments to the Author**

1. Is the manuscript technically sound, and do the data support the conclusions?

Reviewer #1: Yes

Reviewer #2: Partly

Reviewer #3: No

2. Has the statistical analysis been performed appropriately and rigorously? 

Reviewer #1: Yes

Reviewer #2: Yes

Reviewer #3: Yes

3. Have the authors made all data underlying the findings in their manuscript fully available?

Reviewer #1: Yes

Reviewer #2: Yes

Reviewer #3: Yes

4. Is the manuscript presented in an intelligible fashion and written in standard English?

Reviewer #1: Yes

Reviewer #2: Yes

Reviewer #3: Yes

5. Review Comments to the Author

Reviewer #1: The present work aims to investigate attack or escape behaviors in environmental situations that produce emotions of anger or fear in the protagonist. The central role of the study turns out to be to investigate not so much the hedonistic motivations (Hirschman and Holbrook (1982) state that hedonistic motivations are characterized by the presence of a strong emotional component, concerning such aspects as fun and pleasure, rather than the mere achievement of functional goals), but rather, the authors want to focus on the instrumental factors that may influence/modulate anger or fear and attack/escape behaviors.

Specifically, the authors consider the construct of affordances that underlies James J. Gibson's "ecological theory of perception," with the intent of demonstrating its causal relationships between environmental affordances and the expression of anger and fear, whereby more anger is expressed when affordances are available that favor approach and participants choose to fight. In addition, it is intended to show how instrumental motivations influence emotional and behavioral regulation when the environment presents different affordances.

In the study, threats are represented by the presentation of short video clips from horror movies, selected in a previous pilot study, and affordances are presented to study participants in the form of pictograms.

The abstract should be more structured to include: background, objectives, participants, proposed intervention, results, limitations and conclusions.

In the introduction, the authors summarize the main research questions and key findings.

The authors very reductively identify the current literature on the topic and explain how the study relates to this previously published research, but it would be appropriate to expand the review of current scientific literature in the field of studying emotions, affordances, and action motivation. Also, it would be interesting to include some scientific literature regarding the psychology of media?

The introduction section should include citations of the theories and empirical cases cited (lines 12, 23-24 and 38 of the introduction).

In the "experimental procedures" section, it is specified that: "each participant will be made to watch a total number of 40 video clips" (are there any other data that were not reported among the figures in the article? If so, why were they not included?).

The materials and methods section of this study describes the techniques and materials used in the experiments. The explanation of the structure of the scientific study (experimental design used) is missing. This description should be such that another group of researchers can reproduce the experiments in the scientific publication. Consequently, without this section, the reproducibility of the results obtained during the experiment is slightly compromised.

The experiments or interventions are appropriate to address the research question. The conditions are appropriate and the controls right. The data are sufficient to draw a conclusion. The authors do not critically and fully address the limitations of the research. The data were collected and interpreted accurately. The study complies with ethical guidelines.

Writing quality and clarity: writing quality needs to be slightly improved for minor typos.

Bibliographic list: the references in the manuscript appear to be correct.

Conclusions: the article lacks an adequate amount of references to the existing scientific literature on the topic discussed (especially regarding the construct of affordances). The results are extensively argued and discussed.

Weaknesses include the failure to include data from pilot studies, due to technical problems that prevented the article's authors from including them. As pointed out in the "method" section, "participants" paragraph. Can't something be done to solve this problem? (Example: inclusion of a partial report supporting the cited hypothesis). Also missing is reference to the experimental design used in the present empirical study. References to media psychology are missing. The introduction should include a review of broader scientific literature on the constructs investigated (emotions, affordances, motivation)

No major changes to the manuscript are highlighted. Review minor details.

Reviewer #2: "The introduction is generally well-written, but the final paragraph seems to belong in the methods section. Additionally, stating the hypothesis explicitly in the last paragraph would be helpful.

The relationship between the sentence "There is indeed evidence that prior to engaging in conflict, people are more likely to choose situations that are likely to elicit anger rather than neutral emotions (Tamir, 2009)" and the preceding text is not clear.

It would be helpful to explain what HIT is and why this selection criterion was used.

The sentence "Confirming our initial hypothesis, we used design from the pilot study to conduct a power analysis" appears to be worded incorrectly.

It would be beneficial to provide more details on how the power analysis was conducted and why a minimum of 250 participants was necessary.

Could you provide some examples of the plots of the video in the methods section and include them in the appendix?

Has there been research on whether the results may differ if participants report their emotions before reporting behavioral options? In this sequence, emotions may affect behavioral choices, rather than the participants' behavioral choices affecting their reports of emotion.

The current results indicate that if participants see a weapon before choosing to fight, they will report feeling more anger compared with if they did not see the weapon before their decision to fight. The awareness of the affordance to fight increases anger. In contrast, if participants see an escape route before choosing to flee, they will report feeling less fear compared with if the escape route is blocked. The awareness of the opportunity to escape decreases fear. It would be helpful to discuss why this difference in anger and fear exists.

One limitation of the method is that participants were only observing movie clips rather than reporting their own personal emotional experiences - imagining oneself as the movie protagonist and experiencing the events in real life can be very different, especially as horror movie scenes rarely occur in real life. These factors may affect the ecological validity of the study.

The study did not reference recent literature such as Suri, G., Sheppes, G., Young, G., Abraham, D., McRae, K., & Gross, J. J. (2018). Emotion regulation choice: The role of environmental affordances. Cognition and Emotion, 32(5), 963-971.

The discussion should place the study within a larger theoretical framework of "emotion-cognition-behavior-environment" rather than being limited to environmental affordances. In this study, is emotion a kind of cognition, feeling, or motivation? You could refer to “Andrea Scarantino, The Philosophy of Emotions and Its Impact on Affective Science, in Barrett, L. F., Lewis, M., & Haviland-Jones, J. M. (2016). Handbook of emotions, fourth edition. Guilford Publications."

Reviewer #3: In this paper, the Authors try to show a relationship between environmental affordances, decision-making, and emotions in contexts of threat. To this end they adopt an experimental paradigm in which subjects are shown a short horror movie clip after being instructed about the availability of different affordances (either fight or escape) that could help the subject to make a choice between approach behavior (fight through the situation) and avoidance behavior (escape from the situation). The authors conclude by stating that their findings reveal features that items available in the environment (environmental affordances) influence both behaviors (approach or avoidance) as well as levels of experienced anger and fear.

The work proposed by the authors, although interesting, in my opinion presents methodological problems that unfortunately severely hampers the accuracy of the conclusions.

Major points:

My fundamental problem with this study is the experimental approach used; in fact, the study is entirely built using an online approach and thus, the accuracy could only be poorly controlled.

Everything is based on a good understanding by the subjects enrolled in the study, of the structure and purposes of the experiment, but a real control of the accuracy of the answers is completely neglected. Furthermore, despite the large number of subjects enrolled, an accurate selection of subjects is also lacking. In fact, given the subjects of the study, a better characterization of the personality or presence/absence of neuropsychiatric disorders would have guaranteed better control of the experimental conditions and stronger support to the conclusions. Further, since this is a study that calls into question emotions such as fear and anger, I wonder why the authors decided to use an online approach that necessarily prevent any physiological measurement. I think that even the simplest evaluation of a few key autonomic parameter (such as GSR, heart rate, etc.) could have made a great contribution to evaluate the differences in the various experimental situations. In summary, in my opinion the both the recruitment procedures and the lack of control of the experimental procedure, severely hamper the reproducibility of the experiment.

Minor points:

Can the Authors please provide greater justification for this study in the introduction?

In the Discussion the results should be interpreted and discussed in more detail considering the existing literature.

6. PLOS authors have the option to publish the peer review history of their article (what does this mean?). If published, this will include your full peer review and any attached files.

Reviewer #1: **Yes: **Donatella Ciarmoli

Reviewer #2: **Yes: **Chao S. Hu, Southeast University, China

Reviewer #3: No

---

## [Author Response · Author response to Decision Letter 0]

22 Jun 2024

Editor’s Comments

Q1. - how affordances have been operationalized. As far as I understood, "affordances" were written words or maybe icons (judging by figure 1..? this should be clarified) - justification and possibly references should be provided to sustain that those can be properly considered affordances; according to Gibson, affordances are a matter of perception more than cognition; is a written word or a symbolic icon an affordance? To what extent?

A1: Thank you for the opportunity to clarify this. We now explain that affordances were presented in the form of written descriptions on page 4 of the revised manuscript. 

As further explained in our revised manuscript, these are affordances in the sense of Gibson, but also in the way theorists have considered affordances since, expanding on Gibson. Specifically, we now state in page 4, paragraph 2: 

In this paper, we define affordances as the variations in what is available to people to respond to threats they face. This is in keeping with the concept of affordance as originally developed by Gibson and developed by subsequent theorists. Indeed, whilst Gibson’s original work was on perception, affordances were defined as “capturing an object’s value for action in the world” and were defined by Gibson to “cut across the dichotomy of subjective-objective” (Gibson The Theory of Affordances 1979) and can include objects in the environment as much as other people and concepts. In the same vein, affordances have been used later not only in psychology, but also in robotics, design and engineering. In this sense, affordances can themselves be injurious or advantageous (Anelli 2012 https://pubmed.ncbi.nlm.nih.gov/23041720/). In our case, affordances offer different potential action outcomes, ranging from an enhanced ability to fight (e.g. weapon affordance) to an enhanced ability to flee (escape affordance).

Q2 - by using horror movies, this is basically a media psychology research, yet media psychology literature is absent. Immersion and understanding of narrative contents should be taken into consideration - personally I would make different choices if I had to face the mutant monster from "The Thing" or a person possessed by a demon... is a gun effective against both...? Are all weapons the same (e.g., close contact vs. distance)? Research on video games and interactive narrative may be useful to respond

A2; Thank you for bringing up the necessity to introduce more literature to explain and justify our means of using movie clips to study emotions. We agree that there are elements of media psychology in our study, but believe ours to be primarily a study of emotions and therefore to come under affective, rather than media, psychology.

A comprehensive meta-analysis detailed in a paper published by PLOS ONE https://journals.plos.org/plosone/article?id=10.1371/journal.pone.0225040 explores the effectiveness of using film clips to induce positive, negative, and neutral emotional states in research. This study highlights that film clips are particularly effective because they create dynamic contexts with stimuli similar to real-life situations, thus avoiding ethical concerns associated with other emotion manipulation techniques. (In our case, using fantasy horror movie clips also provides the advantage of studying survival responses in a safe and controlled way) The analysis further underscores the standardized nature of film clips in eliciting discrete emotions and maintaining subjective and physiological changes over time. This provides evidence that movie clips can be used in cognitive and social psychology to study emotional expressions. 

We have now included this justification on page 4, paragraph 3 of the revised manuscript. 

We agree with the reviewer that there is a lack of sensitivity in our design to account for the variance of efficacy of affordances in the “weapon” conditions, where one individual might find a particular item more/less useful against specific adversaries shown in the movies. However, in both our initial instructions and in training sessions, the participants were asked to focus on the emotional experience and intuitive reactions towards the threatening situations displayed in the movies, rather than finding a rational solution to wipe out the threats. Therefore, the subjective efficacy of the items will matter less in this scenario. If anything, this unaccounted-for variance would have decreased our ability to detect significant differences between conditions. We further expand on this issue in the study limitations though.

We have now included this as limitations on page 15, paragraph 2 of the revised manuscript. 

Reviewer’s comments: 

Reviewer 1

Q1: The abstract should be more structured to include: background, objectives, participants, proposed intervention, results, limitations and conclusions.

In the introduction, the authors summarize the main research questions and key findings

A1: The abstract has been restructured as follows on page 1 of the revised manuscript: 

In this pre-registered study, we ask how people’s emotional responses under threat may be causally affected by what is available to them in the environment, i.e. environmental affordances. For this purpose, we introduce a novel behavioral paradigm using horror movie stimuli to simulate threats. The study illustrates that affordances, specifically items present in the environment, are instrumental in modulating both behavioral choices (approach or avoidance) and emotional expressions of anger and fear. We found that, approach-related resources, such as possession of a weapon, heightened anger and the propensity to confront the threat. This underscores the influence of environmental affordances on emotional regulation and supports a theoretical framework that connects instrumental motives with the variability of emotional and behavioral responses based on affordances. The research, while innovative, recognizes the constraints of simulated threats and controlled settings, suggesting avenues for future exploration in more naturalistic environments.

Q2: The authors very reductively identify the current literature on the topic and explain how the study relates to this previously published research, but it would be appropriate to expand the review of current scientific literature in the field of studying emotions, affordances, and action motivation. Also, it would be interesting to include some scientific literature regarding the psychology of media?The introduction section should include citations of the theories and empirical cases cited (lines 12, 23-24 and 38 of the introduction)

A2: Thank you very much for this suggestion. Some of the concerns articulated here were also raised above. We now cite a metanalysis on studies of using film clips specifically. We also expand on Gibson and subsequent work about affordances. Please refer to page 4, paragraph 3 of the revised manuscript. 

In this paper, we define affordances as the variations in what is available to people to respond to threats they face. This is in keeping with the concept of affordance as originally developed by Gibson and developed by subsequent theorists. Indeed, whilst Gibson’s original work was on perception, affordances were defined as “capturing an object’s value for action in the world” and were defined by Gibson to “cut across the dichotomy of subjective-objective” (Gibson The Theory of Affordances 1979) and can include objects in the environment as much as other people and concepts. In the same vein, affordances have been used later not only in psychology, but also in robotics, design and engineering. In this sense, affordances can themselves be injurious or advantageous (Anelli 2012 https://pubmed.ncbi.nlm.nih.gov/23041720/). In our case, affordances offer different potential action outcomes, ranging from an enhanced ability to fight (e.g. weapon affordance) to an enhanced ability to flee (escape affordance).

Q3: In the "experimental procedures" section, it is specified that: "each participant will be made to watch a total number of 40 video clips" (are there any other data that were not reported among the figures in the article? If so, why were they not included?).

The materials and methods section of this study describes the techniques and materials used in the experiments. The explanation of the structure of the scientific study (experimental design used) is missing. This description should be such that another group of researchers can reproduce the experiments in the scientific publication. Consequently, without this section, the reproducibility of the results obtained during the experiment is slightly compromised. 

A3: We apologize for any confusion the number 40 caused. As stated in the method section, we have a total number of 20 horror movie clips. However, each of them was presented twice for the affordances, thus a total number of 40 clips for each participant. All data among the figures are reported in the article. A full list of video descriptions are now added in the supplementary section of the revised manuscript. 

We thank the reviewer for pointing out the importance of a clear and replicable method section and have taken further steps to enhance these. We emphasize that we have provided full information in our manuscript thus empowering other scientists to replicate and/or reproduce both our experimental paradigm and data analysis. On manuscript page 5-7, in the “Experimental Procedure” section, we provide a detailed breakdown of the experiment which we have now expanded further; on manuscript page 7-8, we also detailed all the pre-registered hypothesis and corresponding analysis performed. In addition, both our experimental scripts and the data analysis pipelines are shared on the study pre-registration page. Interested readers and researchers will be able to replicate our study and analysis given these tools.

Q4: Weaknesses include the failure to include data from pilot studies, due to technical problems that prevented the article's authors from including them.

A4: We did not include the initial samples (pilot study) in the main text since due to a technical server error we had lost access to the raw data. However, we have the minimally pre-processed pilot data and have now included them in the supplementary materials. As can be seen in the supplementary figures (page 21 and 22 of the revised manuscript), our main conclusion that emotional expression is impacted by the environmental affordances is also strongly supported by the pilot data. These served as the basis for our hypothesis test which we pre-registered and were replicated in the subsequent main experiment. These pilot experiments also included exploratory analysis where we compared effects of affordances across multiple different versions of experiments, in order to decide the most sensitive paradigm to participants’ emotional expressions. 

Reviewer #2: 

Q1 "The introduction is generally well-written, but the final paragraph seems to belong in the methods section. Additionally, stating the hypothesis explicitly in the last paragraph would be helpful. 

The relationship between the sentence "There is indeed evidence that prior to engaging in conflict, people are more likely to choose situations that are likely to elicit anger rather than neutral emotions (Tamir, 2009)" and the preceding text is not clear.

A1: We thank the reviewer for the helpful comment on the clarity of the specified sentences. A list of our full hypothesis can be found from page 7 to page 8 of our manuscript. 

The revised sentence can be found on page 4 of the manuscript. 

Q2: It would be helpful to explain what HIT is and why this selection criterion was used.

A2: We have now included explanation of HIT, and why it is a central metrics in selecting Mturk participants. 

Q3: The sentence "Confirming our initial hypothesis, we used design from the pilot study to conduct a power analysis" appears to be worded incorrectly.

A3: We have now removed the confusing sentence. 

Q4: Could you provide some examples of the plots of the video in the methods section and include them in the appendix?

A4; We have now included a full list of videos descriptions in the supplementary material. They can be found from page 22-24 of the revised manuscript. 

Q5: Has there been research on whether the results may differ if participants report their emotions before reporting behavioral options? In this sequence, emotions may affect behavioral choices, rather than the participants' behavioral choices affecting their reports of emotion. 

A5; We have tested a version of our experiment, where the emotional response is reported before the behavioral option. We did not find significant difference in their emotional expression compared to the current version of experiment, and decided to stick to the current version, which according to participants reports, is more naturalistic. 

Q6: The current results indicate that if participants see a weapon before choosing to fight, they will report feeling more anger compared with if they did not see the weapon before their decision to fight. The awareness of the affordance to fight increases anger. In contrast, if participants see an escape route before choosing to flee, they will report feeling less fear compared with if the escape route is blocked. The awareness of the opportunity to escape decreases fear. It would be helpful to discuss why this difference in anger and fear exists.

A6: We do not have an a priori theory about such a differentiation between fear and anger, hence the following should be understood as post-hoc speculation of the phenomenon we observed. 

Agency, or the sense of control over one's actions and their outcomes, is a critical factor in the experience of anger. Access to a weapon in a threatening scenario might enhance the sense of agency, providing individuals with a means to potentially alter the outcome through their actions, thus potentially heightening the expression of anger. In contrast, fear may be more primal and automatic, requiring less reference to sense of agency for its response. Fear is triggered by the perception of imminent danger and activates a fast, subcortical pathway in the brain that bypasses the higher cognitive processing centers (LeDoux, 1996). This pathway enables rapid response to threats, such as fleeing, freezing, or hiding. The availability of an escape route offers a means to avoid the threat, but does not necessarily increase fear, contrarily it might reduce the uncontrollability and perceived vulnerability to the threat from which fear originates (Armfield 2006). The sense of control provided by the presence of an escape route might reduce the unpredictability associated with the threatening situation, which is a key element in the amplification of fear. Also, the ability to escape forges the subject's appraisal of the situation, from one that might be perceived as overwhelmingly threatening and fearful (no escape) to one that is manageable (escape available).

Also, while fear is a response to immediate threat, anger may require a more complex cognitive appraisal of the situation, including considerations of personal values, norms, and the specifics of the threat. This appraisal process can be influenced by the presence of a weapon, or affordance in general. 

We have now added this section into the supplementary discussion of the revised manuscript, page 24.

Reviewer 3

Q1… In summary, in my opinion the both the recruitment procedures and the lack of control of the experimental procedure, severely hamper the reproducibility of the experiment. (Strongly respond on the bases of the reproduction of the results.)

A1: We beg to differ on this point with the reviewer. We have in fact used extremely judicious recruitment procedures and have been meticulous in designing our controls and enhancing the reproducibility of our results. Specifically:

Large sample and adequate power: we have recruited overall 650 (including pilot) participants, which by any standard in experimental psychology is very large and affords us the power to detect differences. 

Stringent participant selection criteria: we have applied very rigorous criteria

---

## [Editor Report · Decision Letter 1]

2 Jul 2024

PONE-D-23-40401R1Subjective Affective Experience under threat is shaped by environmental affordancesPLOS ONE

Dear Dr. Qi,

Thank you for submitting your manuscript to PLOS ONE. After careful consideration, we feel that it has merit but does not fully meet PLOS ONE’s publication criteria as it currently stands. Therefore, we invite you to submit a revised version of the manuscript that addresses the points raised during the review process.

Authors have provided only partial revisions considering my and Reviewers' comments.  - it was asked to include references and justification about written descriptions considered as affordances; for now Authors added speculation and a reference on dangerous affordances, but they did not provide references to previous research on affordances that could justify their rendition as written descriptions. There is no debate on the fact that affordances could be complex and "mediated" by culture and knowledge (e.g., a mailbox is a physical affordance to put something in, and a mediated affordance to send a message to someone if one is aware that something like a postal system exists), but the problem is whether we can consider descriptions as affordances while they are commonly operationalized as objects one can perceive and act upon - it was asked to include media psychology research by me and one Reviewer; this was not done. Authors added some justification about movies as experimental tools that could generate emotions, which again is well known, but not on narrative media to study action and agency Further revision is necessary that fully responds to Editor and Reviewers' concerns

We look forward to receiving your revised manuscript.

Kind regards,

Stefano Triberti, Ph.D.

Academic Editor

PLOS ONE

---

## [Author Response · Author response to Decision Letter 1]

26 Aug 2024

Editor’s Comments

Q1. - how affordances have been operationalized. As far as I understood, "affordances" were written words or maybe icons (judging by figure 1..? this should be clarified) - justification and possibly references should be provided to sustain that those can be properly considered affordances; according to Gibson, affordances are a matter of perception more than cognition; is a written word or a symbolic icon an affordance? To what extent?

A1: Thank you for the opportunity to clarify this. We now explain that affordances were presented in the form of written descriptions on page 4 of the revised manuscript. 

As further explained in our revised manuscript, these are affordances in the sense of Gibson, but also in the way theorists have considered affordances since, expanding on Gibson. Specifically, we now state in page 4, paragraph 2: 

In this paper, we define affordances as the variations in what is available to people to respond to threats they face. This is in keeping with the concept of affordance as originally developed by Gibson and developed by subsequent theorists. Indeed, whilst Gibson’s original work was on perception, affordances were defined as “capturing an object’s value for action in the world” and were defined by Gibson to “cut across the dichotomy of subjective-objective” (Gibson The Theory of Affordances 1979) and can include objects in the environment as much as other people and concepts. In the same vein, affordances have been used later not only in psychology, but also in robotics, design and engineering. In this sense, affordances can themselves be injurious or advantageous (Anelli 2012 https://pubmed.ncbi.nlm.nih.gov/23041720/). In our case, affordances offer different potential action outcomes, ranging from an enhanced ability to fight (e.g. weapon affordance) to an enhanced ability to flee (escape affordance).

Q2 - by using horror movies, this is basically a media psychology research, yet media psychology literature is absent. Immersion and understanding of narrative contents should be taken into consideration - personally I would make different choices if I had to face the mutant monster from "The Thing" or a person possessed by a demon... is a gun effective against both...? Are all weapons the same (e.g., close contact vs. distance)? Research on video games and interactive narrative may be useful to respond

A2; Thank you for bringing up the necessity to introduce more literature to explain and justify our means of using movie clips to study emotions. We agree that there are elements of media psychology in our study, but believe ours to be primarily a study of emotions and therefore to come under affective, rather than media, psychology. We have now added additional justifications of our method for studying agency, actions and emotions, with literature from Media psychology:

To test these predictions, we developed a novel experimental framework in which each threat stimulus is presented multiple times with different affordances, in the context of horror movie clips. Narrative media, such as movies and stories, are powerful tools for examining action and agency. They provide a rich context for understanding how individuals perceive and engage with their environments (Fleetwood, 2016; Meretoja, 2022). This also isolates causal effects of affordances on subjective affective experience. In our experiments, we manipulated both the resources available to fight (fight affordances) and the means of escape (escape affordances), in the form of written descriptions. Affordances, traditionally understood as opportunities for action provided by the environment, also beyond physical objects to include written descriptions that can evoke emotional responses (Pols,2022; Heft and Kytta,2006; Wokke et al., 2017). We label fight affordances in which a weapon or weapon-like tool are available as the “weapon condition”, as opposed to the “non-weapon condition”, where an available tool is not suitable for fighting. Escape affordances in which escape is more difficult are referred to as the “non-evadable condition” as opposed to the “evadable condition.” Clips from horror movies were used as the stimulus, while participants were asked to rate their emotional responses and decisions under the movie scenarios.

We have now included this justification on page 4, paragraph 3 of the revised manuscript. 

We agree with the reviewer that there is a lack of sensitivity in our design to account for the variance of efficacy of affordances in the “weapon” conditions, where one individual might find a particular item more/less useful against specific adversaries shown in the movies. However, in both our initial instructions and in training sessions, the participants were asked to focus on the emotional experience and intuitive reactions towards the threatening situations displayed in the movies, rather than finding a rational solution to wipe out the threats. Therefore, the subjective efficacy of the items will matter less in this scenario. If anything, this unaccounted-for variance would have decreased our ability to detect significant differences between conditions. We further expand on this issue in the study limitations though.

We have now included this as limitations on page 15, paragraph 2 of the revised manuscript. 

Reviewer’s comments: 

Reviewer 1

Q1: The abstract should be more structured to include: background, objectives, participants, proposed intervention, results, limitations and conclusions.

In the introduction, the authors summarize the main research questions and key findings

A1: The abstract has been restructured as follows on page 1 of the revised manuscript: 

In this pre-registered study, we ask how people’s emotional responses under threat may be causally affected by what is available to them in the environment, i.e. environmental affordances. For this purpose, we introduce a novel behavioral paradigm using horror movie stimuli to simulate threats. The study illustrates that affordances, specifically items present in the environment, are instrumental in modulating both behavioral choices (approach or avoidance) and emotional expressions of anger and fear. We found that, approach-related resources, such as possession of a weapon, heightened anger and the propensity to confront the threat. This underscores the influence of environmental affordances on emotional regulation and supports a theoretical framework that connects instrumental motives with the variability of emotional and behavioral responses based on affordances. The research, while innovative, recognizes the constraints of simulated threats and controlled settings, suggesting avenues for future exploration in more naturalistic environments.

Q2: The authors very reductively identify the current literature on the topic and explain how the study relates to this previously published research, but it would be appropriate to expand the review of current scientific literature in the field of studying emotions, affordances, and action motivation. Also, it would be interesting to include some scientific literature regarding the psychology of media?The introduction section should include citations of the theories and empirical cases cited (lines 12, 23-24 and 38 of the introduction)

A2: Thank you very much for this suggestion. Some of the concerns articulated here were also raised above. We now cite a metanalysis on studies of using film clips specifically. We also expand on Gibson and subsequent work about affordances. Please refer to page 4, paragraph 3 of the revised manuscript. 

In this paper, we define affordances as the variations in what is available to people to respond to threats they face. This is in keeping with the concept of affordance as originally developed by Gibson and developed by subsequent theorists. Indeed, whilst Gibson’s original work was on perception, affordances were defined as “capturing an object’s value for action in the world” and were defined by Gibson to “cut across the dichotomy of subjective-objective” (Gibson The Theory of Affordances 1979) and can include objects in the environment as much as other people and concepts. In the same vein, affordances have been used later not only in psychology, but also in robotics, design and engineering. In this sense, affordances can themselves be injurious or advantageous (Anelli 2012 https://pubmed.ncbi.nlm.nih.gov/23041720/). In our case, affordances offer different potential action outcomes, ranging from an enhanced ability to fight (e.g. weapon affordance) to an enhanced ability to flee (escape affordance).

Q3: In the "experimental procedures" section, it is specified that: "each participant will be made to watch a total number of 40 video clips" (are there any other data that were not reported among the figures in the article? If so, why were they not included?).

The materials and methods section of this study describes the techniques and materials used in the experiments. The explanation of the structure of the scientific study (experimental design used) is missing. This description should be such that another group of researchers can reproduce the experiments in the scientific publication. Consequently, without this section, the reproducibility of the results obtained during the experiment is slightly compromised. 

A3: We apologize for any confusion the number 40 caused. As stated in the method section, we have a total number of 20 horror movie clips. However, each of them was presented twice for the affordances, thus a total number of 40 clips for each participant. All data among the figures are reported in the article. A full list of video descriptions are now added in the supplementary section of the revised manuscript. 

We thank the reviewer for pointing out the importance of a clear and replicable method section and have taken further steps to enhance these. We emphasize that we have provided full information in our manuscript thus empowering other scientists to replicate and/or reproduce both our experimental paradigm and data analysis. On manuscript page 5-7, in the “Experimental Procedure” section, we provide a detailed breakdown of the experiment which we have now expanded further; on manuscript page 7-8, we also detailed all the pre-registered hypothesis and corresponding analysis performed. In addition, both our experimental scripts and the data analysis pipelines are shared on the study pre-registration page. Interested readers and researchers will be able to replicate our study and analysis given these tools.

Q4: Weaknesses include the failure to include data from pilot studies, due to technical problems that prevented the article's authors from including them.

A4: We did not include the initial samples (pilot study) in the main text since due to a technical server error we had lost access to the raw data. However, we have the minimally pre-processed pilot data and have now included them in the supplementary materials. As can be seen in the supplementary figures (page 21 and 22 of the revised manuscript), our main conclusion that emotional expression is impacted by the environmental affordances is also strongly supported by the pilot data. These served as the basis for our hypothesis test which we pre-registered and were replicated in the subsequent main experiment. These pilot experiments also included exploratory analysis where we compared effects of affordances across multiple different versions of experiments, in order to decide the most sensitive paradigm to participants’ emotional expressions. 

Reviewer #2: 

Q1 "The introduction is generally well-written, but the final paragraph seems to belong in the methods section. Additionally, stating the hypothesis explicitly in the last paragraph would be helpful. 

The relationship between the sentence "There is indeed evidence that prior to engaging in conflict, people are more likely to choose situations that are likely to elicit anger rather than neutral emotions (Tamir, 2009)" and the preceding text is not clear.

A1: We thank the reviewer for the helpful comment on the clarity of the specified sentences. A list of our full hypothesis can be found from page 7 to page 8 of our manuscript. 

The revised sentence can be found on page 4 of the manuscript. 

Q2: It would be helpful to explain what HIT is and why this selection criterion was used.

A2: We have now included explanation of HIT, and why it is a central metrics in selecting Mturk participants. 

Q3: The sentence "Confirming our initial hypothesis, we used design from the pilot study to conduct a power analysis" appears to be worded incorrectly.

A3: We have now removed the confusing sentence. 

Q4: Could you provide some examples of the plots of the video in the methods section and include them in the appendix?

A4; We have now included a full list of videos descriptions in the supplementary material. They can be found from page 22-24 of the revised manuscript. 

Q5: Has there been research on whether the results may differ if participants report their emotions before reporting behavioral options? In this sequence, emotions may affect behavioral choices, rather than the participants' behavioral choices affecting their reports of emotion. 

A5; We have tested a version of our experiment, where the emotional response is reported before the behavioral option. We did not find significant difference in their emotional expression compared to the current version of experiment, and decided to stick to the current version, which according to participants reports, is more naturalistic. 

Q6: The current results indicate that if participants see a weapon before choosing to fight, they will report feeling more anger compared with if they did not see the weapon before their decision to fight. The awareness of the affordance to fight increases anger. In contrast, if participants see an escape route before choosing to flee, they will report feeling less fear compared with if the escape route is blocked. The awareness of the opportunity to escape decreases fear. It would be helpful to discuss why this difference in anger and fear exists.

A6: We do not have an a priori theory about such a differentiation between fear and anger, hence the following should be understood as post-hoc speculation of the phenomenon we observed. 

Agency, or the sense of control over one's actions and their outcomes, is a critical factor in the experience of anger. Access to a weapon in a threatening scenario might enhance the sense of agency, providing individuals with a means to potentially alter the outcome through their actions, thus potentially heightening the expression of anger. In contrast, fear may be more primal and automatic, requiring less reference to sense of agency for its response. Fear is triggered by the perception of imminent danger and activates a fast, subcortical pathway in the brain that bypasses the higher cognitive processing centers (LeDoux, 1996). This pathway enables rapid response to threats, such as fleeing, freezing, or hiding. The availability of an escape route offers a means to avoid the threat, but does not necessarily increase fear, contrarily it might reduce the uncontrollability and perceived vulnerability to the threat from which fear originates (Armfield 2006). The sense of control provided by the presence of an escape route might reduce the unpredictability associated with the threatening situation, which is a key element in the amplification of fear. Also, the ability to escape forges the subject's appraisal of the situation, from one that might be perceived as overwhelmingly threatening and fearful (no escape) to one that is manageable (escape available).

Also, while fear is a response to immediate threat, anger may require a more complex cognitive appraisal of the situation, including considerations of personal values, norms, and the specifics of the threat. This appraisal process can be influenced by the presence of a weapon, or affordance in general. 

We have now added this section into the supplementary discussion of the revised manuscript, page 24.

Reviewer 3

Q1… In summary, in my opinion the both the recruitment procedures and the lack of control of the experimental p

---

## [Editor Report · Decision Letter 2]

30 Aug 2024

Subjective Affective Experience under threat is shaped by environmental affordances

PONE-D-23-40401R2

Dear Dr. Qi,

We’re pleased to inform you that your manuscript has been judged scientifically suitable for publication and will be formally accepted for publication once it meets all outstanding technical requirements.

Kind regards,

Stefano Triberti, Ph.D.

Academic Editor

PLOS ONE
---

## [Editor Report · Acceptance letter]

7 Oct 2024

PONE-D-23-40401R2 

PLOS ONE

Dear Dr. Qi, 

I'm pleased to inform you that your manuscript has been deemed suitable for publication in PLOS ONE. Congratulations! Your manuscript is now being handed over to our production team.

Kind regards, 

on behalf of

Prof. Stefano Triberti 

Academic Editor

PLOS ONE